

# Records of ctenophores from South Africa

Mark J. Gibbons[1], Steve H.D. Haddock[2], George I. Matsumoto[2] and Craig Foster[3]

[1] Department of Biodiversity and Conservation Biology, University of the Western Cape, Bellville, Western Cape, South Africa
[2] Monterey Bay Aquarium Research Institute, Moss Landing, CA, United States of America
[3] Sea Change Trust, Cape Town, Western Cape, South Africa

## ABSTRACT

Although ctenophores can be conspicuous components of the plankton in coastal marine ecosystems, only six species have been formally described from around South Africa. Using photographs from local community scientists, we add a further three species (*Cestum veneris, Beroe forskalii*?, *Ocyropsis maculata*?) and six morphospecies to the regional fauna. These additions suggest that South Africa has a ctenophore fauna that is amongst the most diverse, globally; an observation in agreement with information from other taxa. Tips on how community scientists can improve their photographic contributions to understanding ctenophore diversity are provided.

## INTRODUCTION

The warm Agulhas Current flows southward at the edge of the narrow continental shelf along the east coast of South Africa, moving progressively offshore and westward to track the edge of the Agulhas Bank just north of East London (*Lutjeharms, 2006*; Fig. 1). Waters of the Agulhas Current are characterised by their oligotrophic nature, and the biota is of low biomass but high diversity (*Gibbons & Hutchings, 1996*). At the southernmost extremity of the Agulhas Bank, the Agulhas Current retroflects eastward, shedding rings, filaments, and eddies into the South Atlantic (*Lutjeharms, 2006*). Upwelling along the south coast of South Africa is confined to the edge of the continental shelf and to capes and peninsulas: waters are seasonally stratified and productive, and circulation is generally sluggish (*Hutchings et al., 2009*). The west coast of South Africa, from the southern edge of the Agulhas Bank to the border with Namibia is bathed by the cold, northward flowing Benguela Current (*Hutchings et al., 2009*). This area experiences coastal upwelling on a seasonal basis: it is eutrophic and biological communities are characterised by high biomass and low diversity (*Gibbons & Hutchings, 1996*).

Moving around the South African coast from east to west, there is a clear change in the physical environment and this is reflected by a change in the affinities of the marine biota, from tropical, through sub-tropical, to warm and cold temperate biogeographic provinces (*Bustamante & Branch, 1996*). Whilst the diversity of most marine taxa generally declines

Corresponding author
Mark J. Gibbons,
mgibbons@uwc.ac.za

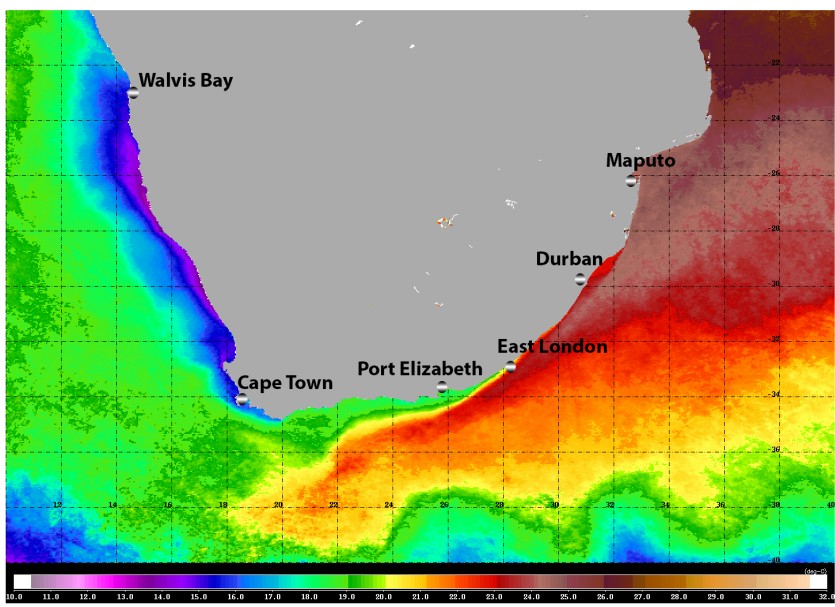

**Figure 1** **Satellite image (4-km resolution MODIS Aqua day-time SST) of the southern African sub-region showing average sea surface temperatures for 2008.** Adapted from *Gibbons et al. (2010b)*, ©2009 Blackwell Publishing Ltd.

from east to west, patterns of endemism indicate that many reach a peak along the south coast (*Emanuel et al., 1992*; *Awad, Griffiths & Turpie, 2002*). Especially in the case of those taxa with restricted dispersal abilities (*Gibbons et al., 2010a*).

Despite its relatively linear coastline of only ∼3,700 km, South Africa supports some of the highest levels of global marine diversity, boasting almost 13,000 species (*Griffiths et al., 2010*). That said, our understanding of much of the region's diversity remains poorly known (*Gibbons et al., 1999a*; *Griffiths et al., 2010*). One taxon that remains particularly knowledge-depauperate is the Ctenophora, despite the fact that members can be conspicuous when present (Fig. 2).

Ctenophores may be found in every ocean basin of the world, from the epipelagic to the bathyal, and whilst most are planktonic others are strictly benthic. They are characterised by a body that comprises approximately 96% water and although they usually possess eight bands of cilia plates (ctene rows), these may be lost in some benthic forms. Ctenophores are predators (*Haddock, 2007*) and tissues will generally contain colloblasts (*Leonardi, Thuesen & Haddock, in press*), which are functionally equivalent to the nematocysts of Cnidaria but rather than sting they stick, ensnaring prey in a glue (*Von Byern, Mills & Flammang, 2010*). Some species possess tentacles with lateral tentillae at some stage in their development, whilst others do not. Almost all species are hermaphroditic (*Harbison & Miller, 1986*).

Ctenophores have gained significant scientific attention in recent years for two main reasons. Firstly, populations (and individuals) can grow fast, and they are known to form blooms (e.g., Fig. 2). *Mnemiopsis leidyi*, a species of pelagic lobate ctenophore naturally found along the east coast of the continental United States was accidentally introduced

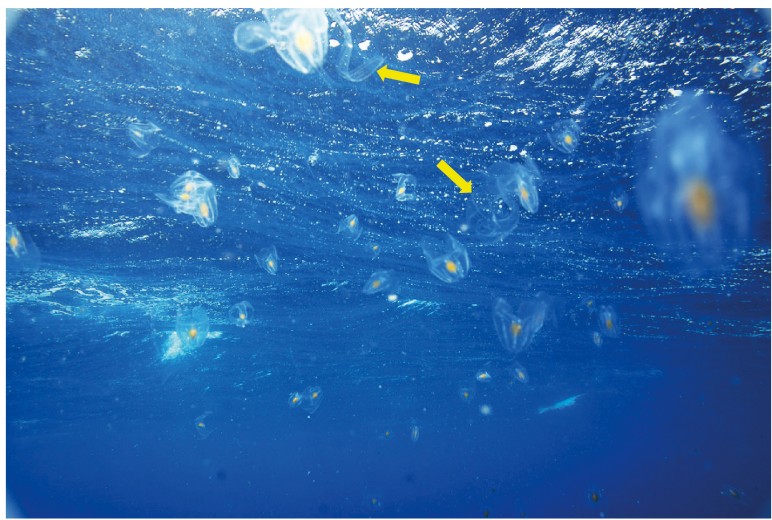

**Figure 2** **An aggregation of *Leucothea* sp. off southern Mozambique: arrows indicate specimens of *Cestum veneris*.** Photograph provided by Jenny Stromvoll.

into the Black Sea during the 1980s (*Shiganova, 1998*). The Black Sea ecosystem at the time was not healthy; stocks of small pelagic fishes had been overexploited and there was heavy eutrophication, which, when combined with a period of warm weather, led to a massive outbreak of these ctenophores that served to restructure the way energy and materials flowed through the ecosystem (*Kideys, 2002*). The Black Sea system has partly recovered following the introduction of another invasive ctenophore, *Beroe ovata* (which preys on ctenophores), whose arrival was accompanied by a decline in eutrophication and a "cold-snap" (*Kideys, 2002*; *Shiganova et al., 2001*). However, *Mnemiopsis leidyi* has subsequently been introduced into the Mediterranean Sea (*Fuentes et al., 2009*) and the Baltic Sea (*Javidpour, Sommer & Shiganova, 2006*), the latter from a separate origin (*Reusch et al., 2010*).

The other reason that ctenophores have made headlines is that they may form a sister taxon to all other metazoans, being considered by some as sister to the Porifera (*Dunn et al., 2008*; *Dunn, Leys & Haddock, 2015*; *Shen, Hittinger & Rokas, 2017*, but see *Simion et al., 2017*). Whatever their phylogenetic position, their affinities with Cnidarians are clearly convergent and not from shared ancestry (*Moroz, 2015*).

Though nine orders, 27 families, and between 150 and 250 species are recognised at present, the higher systematics of Ctenophora is considered a mess (*Giribet & Edgecombe, 2020*). Part of the reason for this must undoubtedly lie with the fact that good specimens on which to base a robust phylogeny are missing. Ctenophores are very delicate animals that "disintegrate before your eyes" in most fixatives, which makes the preservation of definitive material extremely difficult. High molecular weight DNA and good quality RNA can be challenging to extract, and because of their distinct genetics, primers for amplifying standard barcoding primers like COI often fail to work when applied to ctenophores.

In their reviews of what is known about marine diversity around South Africa, *Gibbons et al. (1999a)* and *Griffiths et al. (2010)* recognised eleven species of Ctenophora. These numbers were generated by our late colleague Hermes Mianzan from Mar del Plata in Argentina and followed his treatment of the phylum in *Boltovskoy*'s *(1999)* encyclopaedic work "Zooplankton of the South Atlantic" (*Mianzan, 1999*). Unfortunately, the list of species that Mianzan used in his contribution to *Gibbons et al. (1999a)* has been lost, and published papers on the phylum from the region are largely lacking. Indeed, with the exception of *Moser*'s *(1903)* work on the ctenophores collected during the Siboga Expedition, or her efforts based on the material from the Deutschen Sudpolar Expedition (*Moser, 1910*), there are no formal descriptions of any species from South Africa. References to species in regional ecological works (e.g., *Gibbons, Stuart & Verheye, 1992*; *Gibbons et al., 1999b*; *Gibbons & Buecher, 2001*; *Gibbons, Buecher & Thibault-Botha, 2003*; *Buecher & Gibbons, 2000*; *Gibbons & Painting, 1992*) are not based on definitive identifications, and caution should be exercised in their too literal use beyond the level of genus.

In ecological studies, estimates of ctenophore abundance (counts, volume, biomass) based on material that has been identified to the level of genus are unlikely to be problematic owing to the fact that most members of the same genus will likely play a similar functional role within the ecosystems of which they are a part. In this context, the challenge is not identifying functional groups, but rather obtaining suitable quantitative samples that are in a condition to be enumerated and assessed. The concept of morphospecies is therefore appropriate in such work, and this lends itself to garnering the support of community science. We define a morphospecies here as a morphologically distinct species, which we cannot assign to a formal description, for a variety of reasons. In cases where a range-restricted species is immediately identifiable by its morphology (e.g., the okapi, *Okapia johnstoni*), the morphospecies and the true species are the same. However, in other cases, a morphospecies may comprise a number of true species that are morphologically indistinguishable, or at least difficult to tell apart at a superficial level. This becomes especially pertinent when the morphospecies is widely distributed. For example, jellyfish of the genus *Aurelia* occur in many coastal regions of the world, and for decades the number of taxonomically recognised species was few: almost all specimens recovered in temperate boreal waters were regarded as *Aurelia aurita* (e.g., *Kramp, 1961*). However, it is now understood that *Aurelia aurita* is actually confined to NW Europe (*Jarms & Morandini, 2019*) and that previous records in (e.g.) the Mediterranean Sea, are of other, near-cryptic species (*Scorrano et al., 2017*). Yet from an ecological point of view it generally doesn't matter, because the different true species of *Aurelia* within a morphospecies will likely share the majority of their traits. So it is with the majority of ctenophores.

With the advent of digital photography and the provision of a number of online portals (e.g., iNaturalist; http://www.inaturalist.org or Jellywatch; http://www.jellywatch.org) that allow image sharing, reliable information on diversity is becoming increasingly part of mainstream science (*Silvertown, 2009*). Digital cameras are becoming more affordable, and phones are ubiquitous and capable of taking high-quality geolocated images. Use of these technologies by interested members of the public is becoming widespread as community members share their passions for the environment with others, and get

feedback from experts. For large and charismatic taxa and for taxa that are easily identified morphologically, these images can contribute species-level information (*Falk et al., 2019*; *Kobori et al., 2015*). However, for those organisms that require very detailed images taken under perfect conditions, the information that can be obtained may be of value only at a higher level of identification.

As noted previously, morphospecies can be used to assess trends and patterns both in geographic distribution and temporal cycles (but see below). Here, we aim to do two things. Firstly, we use images provided by a variety of community scientists to update our knowledge of ctenophores around South Africa, as *Deidun (2011)* did around Malta. Given what we know about the diversity of other marine taxa in the coastal waters around South Africa (*Griffiths et al., 2010*), we hypothesise that taxonomic richness will be high. Secondly, owing to the generally inaccessible nature of the key literature in a developing region like southern Africa, we highlight important features that should allow identification of specimens in the field, and we provide comments about similar species. In the hope that future photographs will be more valuable, details that would permit possible identification to species level are also given, as is a brief overview of ctenophore classification and anatomy.

## Overview of ctenophore classification and anatomy

Ctenophores either have tentacles at some stage in their life (Class Tentaculata) or they lack them (Class Nuda), though it should be appreciated that many tentaculate species may lose them in adulthood, or they may become very much reduced. A good source for valid ctenophore species names is the web site of *Mills (1998-present)*. The Class Nuda contains only one Order (Beroida), a single Family (Beroidae) and two genera, typified by *Beroe*, which is the most speciose. The Class Tentaculata is divided into eight orders, which can be distinguished by their benthic (Platyctenida) or pelagic habit, whether they possess oral lobes (Lobata) or not, as well as their general shape: approximately spherical (some Cydippida), elliptical (some Cydippida), flattened (Cestidae) etc. Many of the remaining orders are either bathypelagic and thus beyond the scope of community science, or are very rare and unlikely to be encountered by community scientists; they are ignored here.

When it comes to describing ctenophores, as with all things, it is important to get your bearings: what is up, what is down; what is left and what is right. This is especially important for organisms like ctenophores, which show biradial symmetry. There are two "ends" to a ctenophore: an oral end with a mouth and an aboral end with an often near-invisible sense organ called a statocyst. Regardless of the taxon, these "ends" can always be located in the mid-line of the animal. Food is ingested via the mouth and enters a flattened pharynx (stomodaeum) where digestion takes place, before being distributed through a series of canals and exiting via excretory pores (also near-invisible to the naked eye). If you were to look at a ctenophore from the side (aboral-end up, oral-end down), and you turn it around through 360° you will have two "full frontal" views of the stomodaeum (not one "front" and one "back"—remember the rotational symmetry), and two "side-on" views (Fig. 3). Note that these views do not have exact bilateral or mirror symmetry, meaning that the left half of a front view may be slightly different than the right half. When you view an oral

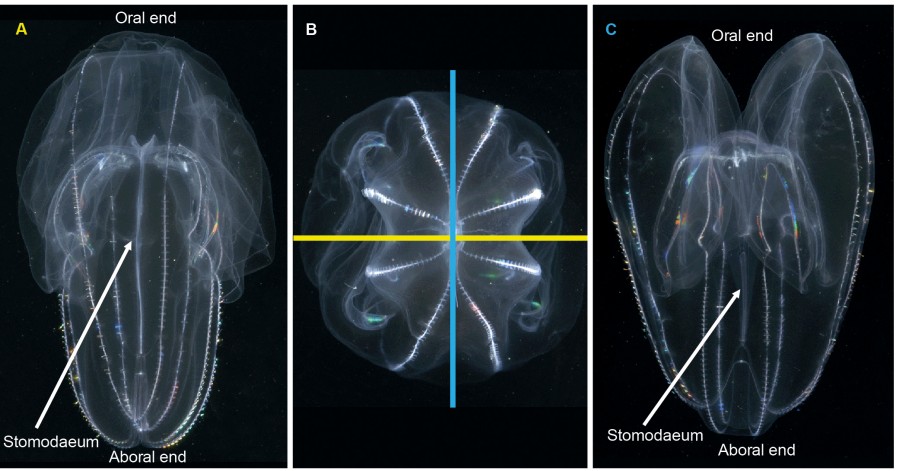

**Figure 3** **Symmetry planes of a ctenophore, using _Bolinopsis vitrea_ as model.** (A) view of the tentacular plane; (B) aboral view, showing the stomodaeal axis (in blue) and the tentacular axis (in yellow); and (C) view of the stomodaeal plane. (photos: A. Migotto) Reproduced with permission of O. Oliveira; adapted from _Oliveira et al. (2007)._

or aboral image of a ctenophore you can see both the stomodaeal (or pharyngeal) plane of view (blue line in Fig. 3B) and the tentacular plane of view (yellow line in Fig. 3B). This "top-down" perspective is also where the subtle asymmetries are most apparent, in the path connecting to the anal pores.

Pelagic ctenophores will have eight rows of symmetrically arranged cilia plates that are used for locomotion and, with the exception of beroids and adult _Ocyropsis_, two tentacles of variable length and arrangement (Fig. 4). The four ctene rows adjacent to the stomodaeum and the stomodaeal plane are referred to as sub-stomodaeal, whilst those adjacent to the tentacles and the tentacular plane are known as sub-tentacular –and the two sets may be of different lengths. The numbers of cilia per plate and the number and spacing of plates per row can be important features at the species-level. A pair of extensible tentacles retract to a bulb, and emerge from the body of Cydippida via tentacular sheaths (Fig. 4A). Where the tentacles emerge along the body is important for identification purposes, as too is information on the relative size and internal orientation of the sheaths. Most tentacles are deployed angled away from the mouth, but in a few genera (_Lampea_, _Dryodora_) they may emerge laterally, or even be orally directed as in _Haeckelia_. Tentacles may possess side branches called tentilla; or they may not: this is also important for identification. On these tentilla are the specialized colloblasts, which aid in prey capture.

Lobate ctenophores possess two cup-shaped oral lobes that are obvious viewed from the stomodeal plane, and four auricles (Fig. 4B). The relative shape, size and thickness of the lobes are important features for identification purposes, as too is the arrangement of the various canals that run through the tissue. External papillae may occur in some taxa. The relative length of the auricles, which are situated at the base of the lobes near the mouth, varies between taxa; they may be coiled (or not) and slender (or not) and have ciliated

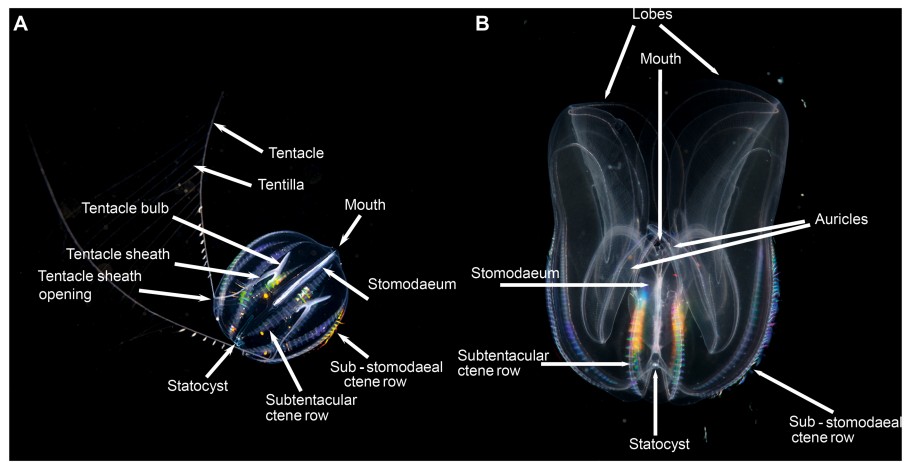

**Figure 4** Basic ctenophore anatomy, (A) *Pleurobrachia pileus*, in stomodaeal plane, as an example of a cydippid ctenophore; (B) *Mnemiopsis leidyi*, in stomodaeal plane, as an example of a lobate ctenophore. (A) photograph by Kåre Telnes, http://www.seawater.no/fauna/ctenophora/pileus.html, (B) photograph by Bruno C. Vellutini, CC BY-SA 3.0, https://commons.wikimedia.org/w/index.php?curid=30155106.

edges. It should be realized that juvenile lobate ctenophores possess long tentacles and lack obvious lobes, and that they resemble cydippids.

Although pelagic ctenophores are generally translucent, they may possess pigment spots and can be bioluminescent. The latter is challenging to video or photograph, but should be noted.

## Taking pictures of ctenophores in situ

The vast majority of recreational divers and underwater photographers (community scientists) rarely have the time, tools, facilities, or collecting permits to capture animals, transport them to an aquarium and photograph them ex situ. Consequently, we do not summarise the methods that would be used to take such images here, and the interested reader is referred to the internet where a number of resources about aquarium photography can be found (e.g., *Haddock, 2011*).

All of the images used here were captured in situ. Good pictures require sunny conditions or the use of lights. The subject should ideally be backlit by the sun, and the background should be dark. It often helps to underexpose by up to two stops to bring out details and not to overexpose bright areas. Be cautious not to swirl water near animal or disturb its rhythm. Try to photograph the animal from six different angles to get all sides of it. It can be useful to do a few shots directly as well as from below for a different kind of translucent detail. Avoid blurring by keeping the camera still and gently depressing trigger. Take many pictures and edit the clearest ones. Take wide-angle, close ups, and macro shots if possible. Relax the hands and body while working. Animals near the surface will normally have better light. Use backlighting with a separate light if animals are deeper.

### The images

Although iNaturalist represents an extremely valuable resource for studying morphospecies, and we encourage all interested community scientists to upload their images there or to another similarly organised platform, the coverage of ctenophores is incomplete. At the time of writing, there are a total of sixteen ctenophore records on iNaturalist for South Africa, which is clearly not enough to report on regional biodiversity! As a consequence, a personal request was made to the network of known underwater photographers in the region for them to dig deep into their archives and send through any captured images. The photographs used here, therefore, do not represent the total population of possible images but simply those from the sampled photographers. Full details of contributors can be found in the acknowledgements and, where appropriate, in the Figure legends. With the exception of the images of *Lyrocteis* sp., all the photographs shown here were collected by divers using SCUBA or snorkel gear with a variety of underwater cameras. Some of the pictures were taken in water with a depth shallower than 3 m, whilst others were "snapped" during decompression stops after deep dives: all were taken close to the surface.

## THE MORPHOSPECIES

### Tentaculata
### *Cydippida*

*Hormiphora* sp. Morphospecies. Pelagic. Body a prolate spheroid, slightly broader at oral end, may be moderately compressed in the tentacular plane, up to 30 mm in length, transparent; tentacle bulbs in close proximity to stomodeum, between 25–50% stomodeum length; tentacle sheaths parallel stomodeum for some distance; stomodeum greater than 50% body length; all eight ctene rows approximately equal in length, extend greater than 75% of body length from aboral end. Seventeen species recognised globally, and although none have formally been described from South Africa it is known to occur from northern KwaZulu Natal to the SW Cape (Figs. 5A and 5C). Good pictures require a focus on the length and position of the tentacle bulbs relative to the stomodaeum, where the tentacle exits the body, detail of tentacle and any unusual shape to the tentilla (coiled, "hand-shaped", globular), as well as the overall body shape.

    *Pleurobrachia* sp. Morphospecies. Pelagic. Body approximately spherical, less than 20 mm in diameter, transparent; tentacles emerging close to aboral end, tentacle bulbs short, less than 25% stomodeum (stomodaeum) length; tentacle sheaths orientated at pronounced angle from stomodeum; stomodeum less than half body length. All eight ctene rows approximately equal in length, extend 75% of body length from aboral end. Ten species recognised globally, two of which have been described from South Africa: *Pleurobrachia pileus* from the west coast and *P. pigmentata* from Durban (*Moser, 1903*; *Moser, 1910*). Characters that can be used to distinguish between these two species include, in the case of *P. pigmentata*, pigmentation along the ctene rows, very wide dense ctene rows, and what is described as a constriction of the oral third of the body. A shallow water taxon, species of *Pleurobrachia* (sea-gooseberry) may be common in harbours and enclosed embayments around the coast (Figs. 5B and 5D). Good pictures require a focus on the

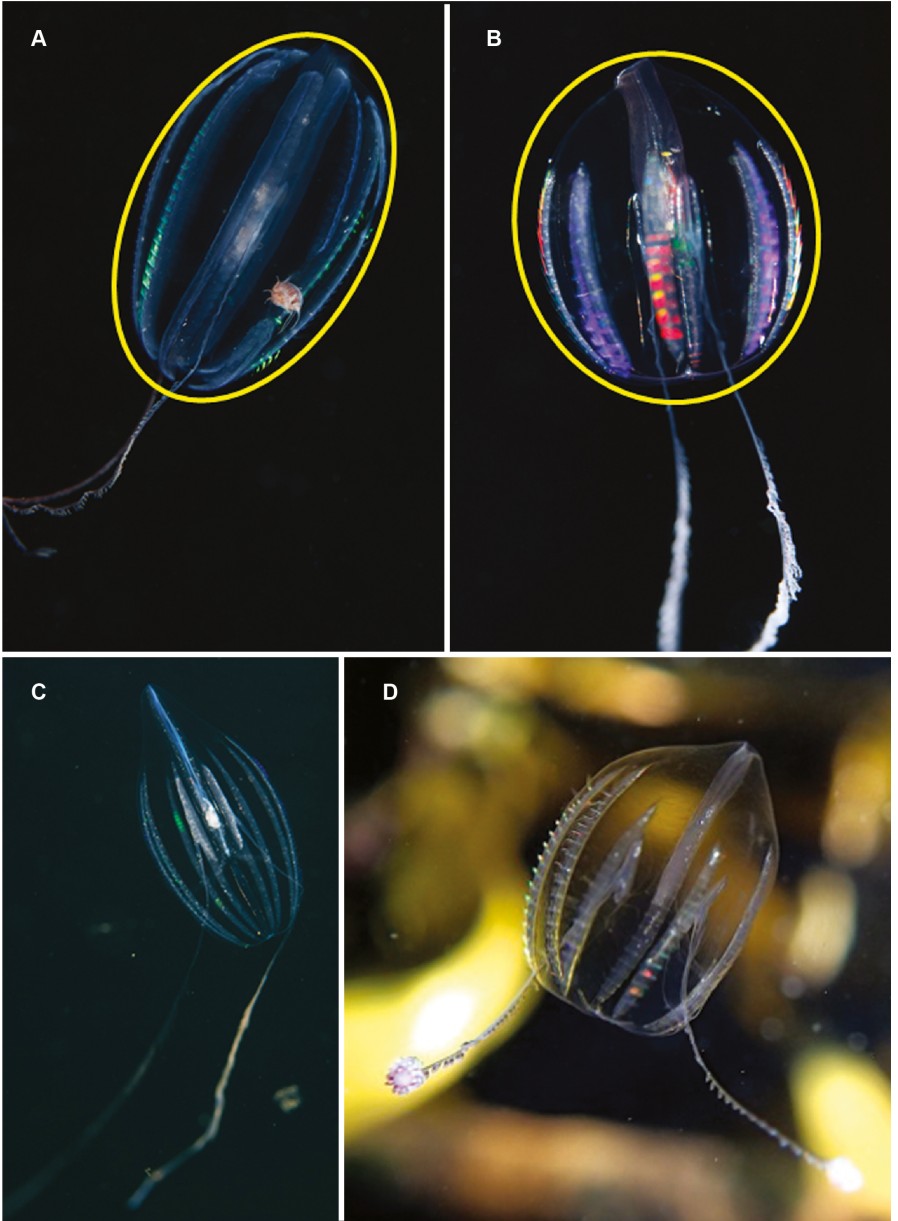

**Figure 5** **Differences in overall body shape of the cydippid ctenophores *Hormiphora* sp. (A, C) and *Pleurobrachia* sp. (B, D).** Photographs provided by: (A, B) Guido Zsilavecz, (C) Georgina Jones, (D) Craig Foster; taken at Oudekraal (A, B) along the west coast of the Cape Peninsula, (C) in Sodwana Bay in northern KwaZulu Natal and (D) in False Bay. Note the hyperiid amphipod associated with *Hormiphora* sp. (A); such parasitic associations with ctenophores are not uncommon.

origin and angle of the tentacular sheaths relative to the stomodeum, the relative lengths of the ctene rows and the density of ctenes. It should be stressed that many tentaculate ctenophores have larvae that resemble *Pleurobrachia*, so caution should be exercised in putting names to small cydippids.

*Callianira antarctica*. Species. Pelagic. Although generally regarded as a polar species, *Moser (1910)* collected two specimens in oceanic waters NW of Cape Town at a temperature of 14.6 °C. Body strongly compressed in the stomodaeal plane, ovoid but with two aborally projecting wing-shaped keels, giving it an arrowhead shape; transparent; to 16 cm. Tentacles emerge, and all ctene rows end, at base of keels. Sub-stomodeal ctene rows slightly longer than sub-tentacular ones. Unlikely to be seen close to continental South Africa. Not illustrated.

### Cestida

*Cestum veneris* . Species. Pelagic. Body laterally expanded and flattened, long and ribbon-like, up to 1.5 m in "length" (=width), transparent. Tentacles fringe the mouth opening, which extends along the "length" of oral surface, with tentilla forming a veil across body. Four (sub-stomodaeal) ctene rows run along "length" of aboral surface, two each side; four short sub-tentacular ctene rows around sense organ. "The set of canals that run along the middle of the body originate near the base of the stomodeum, rapidly curving up to the midline" (*Mills & Haddock, 2007*). In other words, if you trace the canal running along the midline from a "wingtip" toward the center of the body, it will make an S-shaped digression before joining the canals at the center of the body. Contrast this with *Velamen parallelum*, whose canals continue straight from the tip to their point of intersection. *Velamen* does not reach the sizes of *Cestum*, but it may be hard to distinguish small specimens. Other keys to look for include: *Cestum* may have purplish-black pigment on its wingtips, and the gonads of *Velamen* form a frosty-looking dashed line vs. a continuous line in *Cestum; Cestum* can also roll itself up while *Velamen* cannot. Both species are monotypic and occur worldwide in tropical and subtropical regions, moving laterally in open water by body undulations (slow undulations in *Cestum;* rapid wriggling in *Velamen*). Recorded around South Africa from southern Mozambique to False Bay, Cape Town (Fig. 6). Distinguishing the two species requires good photographs of the stomodaeal region to highlight the presence/absence of sub-tentacular ctene rows and the origins of the lateral canals.

### Lobata

*Ocyropsis* sp. Morphospecies, but see below. Pelagic. Body resembles "the general shape of two hands held together in prayer" (*Gershwin, Zeidler & Davie, 2010*), characterised by the presence of two large, lateral muscular lobes that extend well beyond the mouth; generally less than 5 cm in "length"; pale and translucent, lobes sometimes bearing black or brown pigment spots. Although the latter feature is diagnostic for *Ocyropsis maculata* (arrow in Fig. 7B), we need to remember that incomplete global sampling means that local specimens may yet be distinct (hence our use of "?"). The shape of the stomodeum is an important diagnostic feature, as too is the origin of the different ctene rows: sub-tentacular ctene rows shorter than sub-stomodaeal ctene rows, former extending to base of lobes, latter to a variable distance along lobes. Tentacles reduced or absent in adults; with four auricles of variable shape and length (arrow in Fig. 7A), usually appear as a narrow triangle. Although five species are recognised globally from both warm and cold temperate oceans, none have been formally described from South Africa though photographs indicate it occurs in the region from southern Mozambique to False Bay (Fig. 7). Swimming behaviour (a

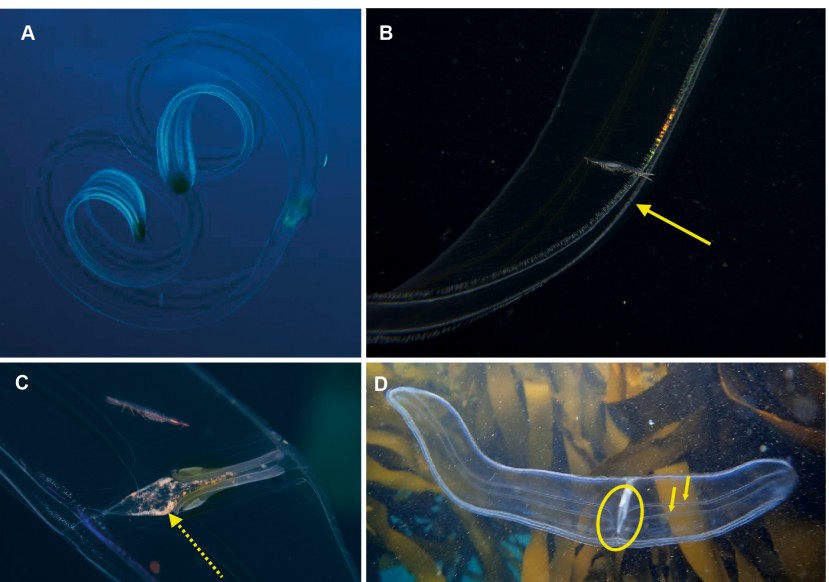

**Figure 6** *Cestum veneris,* **also known as the Venus girdle.** This strap-like ctenophore is characterised by ctene rows (arrows, B) that run along the aboral surfaces and tentacles that run along the oral edges, with tentillae that drape across the body (faintly visible in B and C). A knowledge of the origins (encircled, D) of the lateral canals (arrows, D) is important in distinguishing this species from *Velamen parallelum*. Photographs provided by: (A) Mike and Val Fraser, (B) Jenny Stromvoll, (C) Georgina Jones, (D) Craig Foster; (A–C) in southern Mozambique and (D) in False Bay. Note the hyperiid amphipod (*Streetsia* sp) in (B, C) and what looks like a juvenile euphausiid in the stomodeum (dashed arrow, C).

series of 'claps') is diagnostic. The species can be separated with some difficulty and good pictures require a focus on the ctene rows, stomodaeum, any tentacular apparatus as well as pigmentation (or lack of) on lobes.

*Bolinopsis* sp. Morphospecies. Pelagic. Body egg-shaped, slightly broader at oral end; less than 10 cm in length (including lobes); transparent. Two lobes of variable conspicuity arise between mouth and statocyst to surround and extend beyond mouth a variable distance. Four sub-stomodaeal ctene rows extend a variable distance along lobes from aboral pole, whilst each sub-tentacular row generally terminates at an auricle. Auricles narrow and ribbon-like, reaching to a variable distance beyond mouth. With two short tentacles arising adjacent to, and fringing, oral surface. Ten species recognised globally, and although none have formally been described from the region, it has been photographed around South Africa and appears to extend from southern Mozambique to False Bay (Fig. 8). Some species of *Bolinopsis* are strongly pigmented, others may bear tubercles on outer surface; the arrangement of the canals in the lobes is diagnostic. Good pictures require a focus on the point of connection from the lobes to the body (whether near the level of the mouth or closer to the statocyst) and the shape of the auricles.

*Leucothea* sp. Morphospecies. Pelagic. Body approximately oblong, compressed, with numerous tubercles on outer surface; to a length of approximately 20 cm; often lightly pigmented, transparent. Two prominent lobes arise near the level of the mouth to surround and extend beyond mouth a variable distance. (Lobes coil and change disposition when

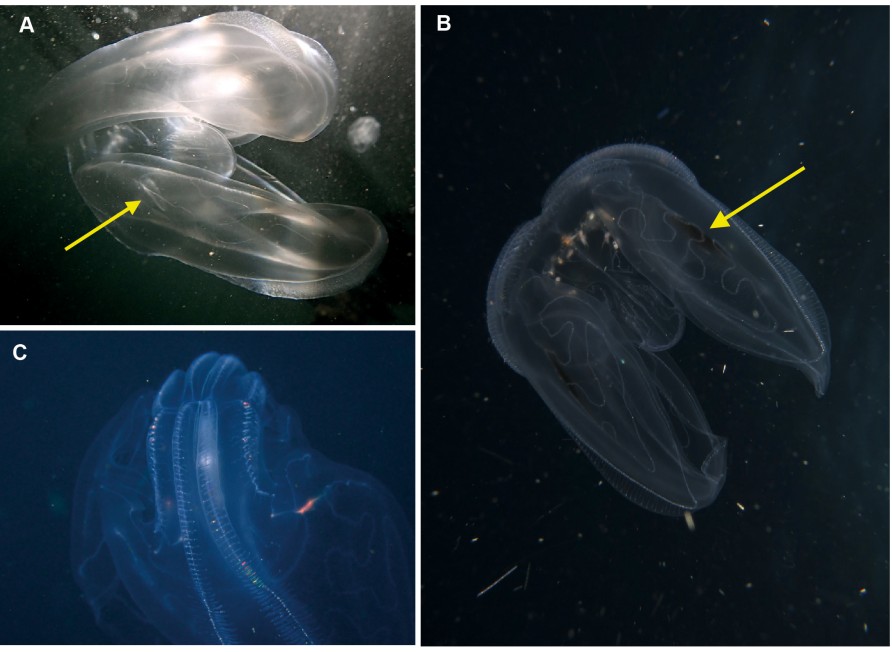

**Figure 7** *Ocyropsis.* **sp (A, C),** *Ocyropsis maculata***? (B), illustrating its "cupped-hands" appearance.** Note the pigmented spots at the base of the lobes of *O. maculata*?. Arrows highlight the auricles (a) and black spot (b). Photographs provided by: (A) Craig Foster, (B) Jenny Stromvoll, (C) Peter Southwood; taken in False Bay (A, C), and southern Mozambique (B).

disturbed.) Aboral end markedly indented. Four sub-stomodaeal ctene rows extend a variable distance along lobes from aboral pole, whilst each sub-tentacular row generally terminates close to the origin of the auricle. Auricles cylindrical or ribbon-like, often coiled, and elongated relative to other lobates. Two long, trailing tentacles arising adjacent to mouth. Seven species recognised globally from tropical and subtropical waters, which can be differentiated by the arrangement of the canals in the lobes, and the nature of papillae, including the presence of orange pigmentation (*Matsumoto, 1988*). Species belonging to this genus have been photographed in the region from southern Mozambique to False Bay (Fig. 9). Good pictures require a focus on the body texture and papillae, the origin of the lobes, and the auricles.

*Eurhamphaea* sp. Morphospecies. Pelagic. Body narrow, compressed in stomodaeal plane, with two relatively firm gelatinous projections aborally that each terminate in a simple filament of variable length to give the animal a pointed, angular appearance; to 10 cm in length, more commonly <5 cm; transparent, with conspicuous red "spots" even as juveniles. Spots may disappear if the organism is disturbed and releases its yellow ink. Statocyst sunk in a deep cleft. With two relatively short lobes and stiff, narrow auricles. Four sub-tentacular ctene rows extend along length of aboral projections; four sub-stomodaeal ctene rows arise at edge of aboral cleft and extend orally to base of lobes. Five species are recognised globally, all of which were first described during the early 1800s, but the

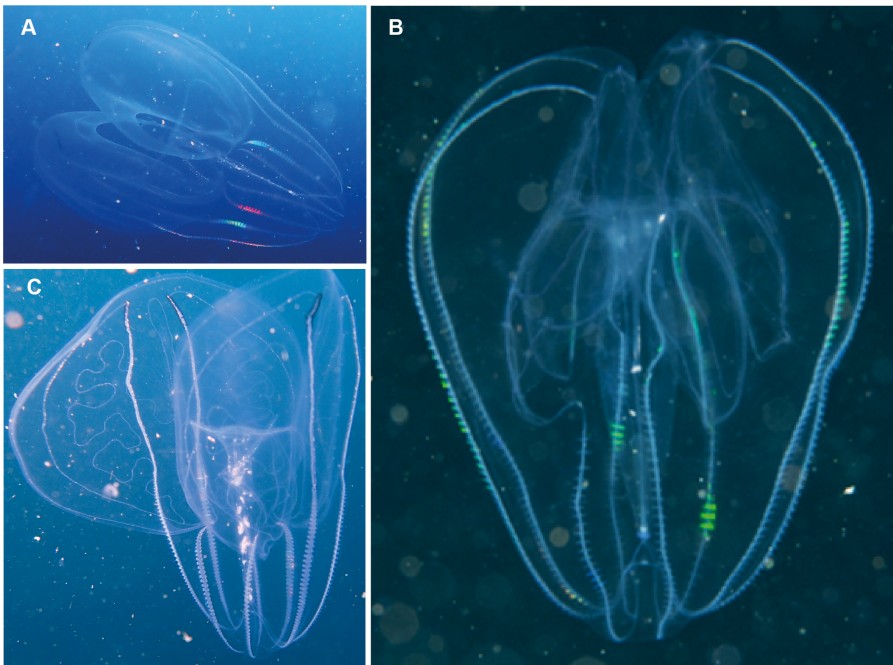

**Figure 8** **_Bolinopsis_ sp.** Photographs provided by: (A, C) Peter Southwood, (B) Craig Foster; all taken in False Bay.

genus is functionally monospecific, with few records of all but *Eurhamphaea vexilligera*. Photographed in the region only from southern Mozambique (Fig. 10).

### *Platyctenida*

*Coeloplana* sp. Morphospecies. Benthic. Body extremely flattened, resembling a free-living flatworm; variously coloured, often brightly but may also be cryptic; rarely more than 30 mm in length. With two, long finely branched tentacles that arise from "chimneys" at variable positions on body. Body surface may have variable numbers of extendable papillae, arranged in variable ways. Lacking ctene rows. Thirty-two species recognised globally; none formally described from South Africa but specimens have been photographed in both southern Mozambique and False Bay (Fig. 11). Very difficult to identify from photographs alone, requiring a focus on coloration and the number and distribution of papillae. They tend to have close associations with other benthic organisms, and the identify of their "host" can assist in identification (e.g., *Matsumoto, 1999*).

*Lyrocteis* sp. Morphospecies. Benthic. Body erect and "lyre-shaped"; sessile; maybe brightly coloured; to a height of 15 cm. Ctenes absent in the adult; a tentacle arises from a furrow at the tip of each aboral arm, tentillae arising on one side only. Arms may (*L. imperator*), or may not (*L. flavopallidus*; *Robilliard & Dayton, 1972*), have longitudinal ridges on outer margin; trunk and proximal part of arms may bear numerous small papillae (*L. imperator*). Two species recognised globally from Japan (*L. imperator*) and Antarctica (*L. flavopallidus*). Recorded locally only in the deep canyons off northern KwaZulu Natal

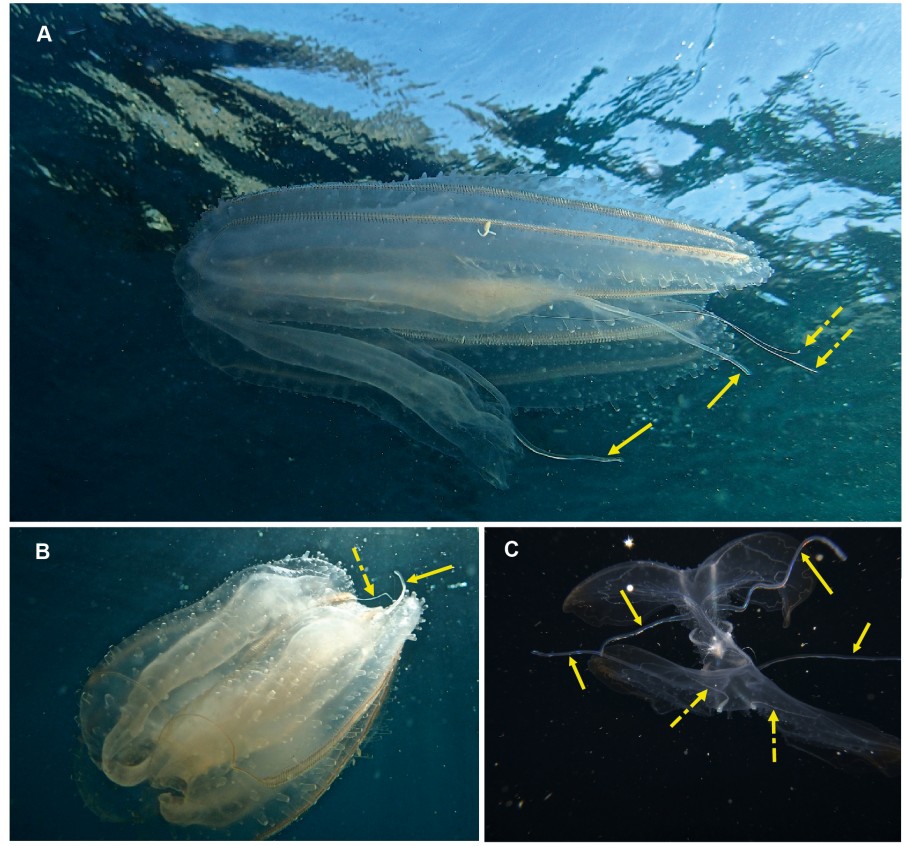

**Figure 9** *Leucothea* **sp., which is characterised by the presence of numerous papillae covering the relatively long body.** Solid arrows highlight the long auricles and dashed arrows indicate the long trailing tentacles. Photographs provided by: (A, B) Craig Foster, (C) Jenny Stromvoll; (A, B) taken in False Bay, (C) in southern Mozambique.

(Fig. 12). Note the variety of colours shown by the specimens photographed, which, given that colour tends to be unique to species of *Coeloplana,* means it is possible that more than one species may be present.

## Nuda
### Beroida

*Beroe* sp. Morphospecies but see below. Pelagic. Body cylindrical and cigar-shaped, variously flattened; opaque, to a maximum of 20 cm. Without tentacles. With eight ctene rows, of variable length. Twenty-six species recognised globally from all oceans and seas, and three species have been formally described from South Africa (*Beroe cucumis, Beroe hyalina, Beroe* (*Pandora*) *mitrata*) according to *Moser (1910)*. Here we add another potential species to the regional fauna, *Beroe forskalii*? (Fig. 13D), which can be identified by its conical shape and generally pinkish hue. The different species of *Beroe* can be differentiated by their shape, the relative lengths of the ctene rows and the arrangement and level of anastomoses of the different canals (Fig. 13C). In *B. forskalii*, side branches arising from the canals connect into a fully connected mesh (anastomose). In other species,

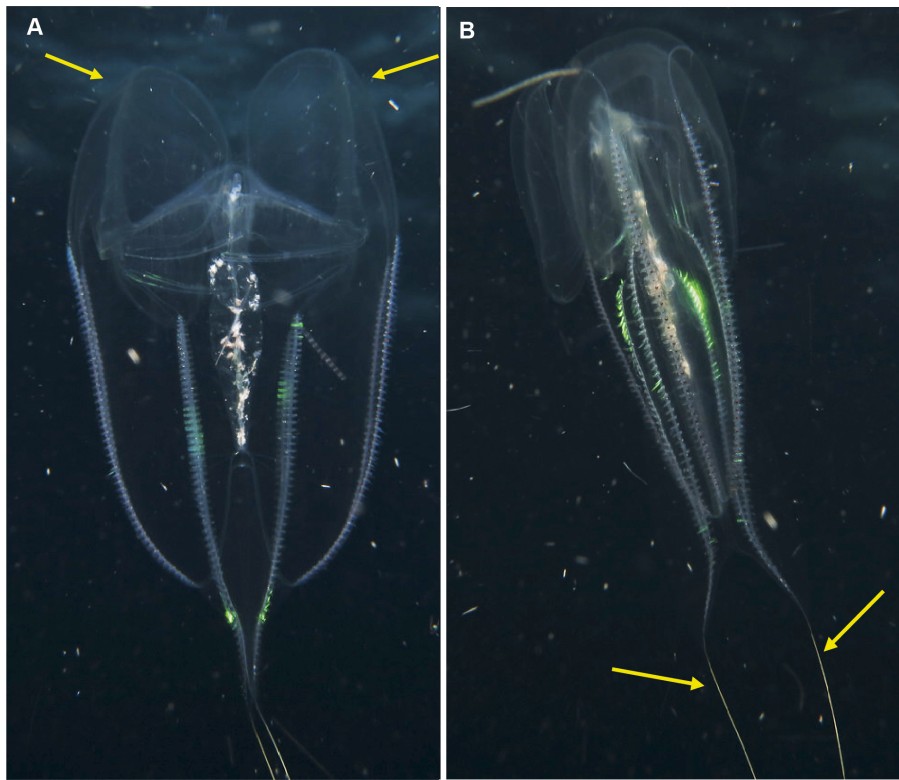

**Figure 10 *Eurhamphea* sp.** An uncommon lobate ctenophore that is characterised by its short oral lobes (arrows, A), aboral projections that each terminate in a simple filament (arrows, B) and prominent "ink" spots. Photographs provided by Jenny Stromvoll; both taken of the same specimen in southern Mozambique.

the side branches may divide multiple time, but they terminate blindly. Good pictures require a focus on body shape, coloration if any, canal structure within body and the shape and form of the oral end of the body. Probably the most common morphospecies of ctenophore recorded in the region, which can be found around the coast from southern Mozambique to the Orange River.

## Comments

As noted earlier, six species of ctenophore have formally been documented and described from around South Africa (*Pleurobrachia pileus*, *P. pigmentata*, *Callianira antarctica*, *Beroe cucumis*, *B. hyalina*, *B. mitrata*), and we have updated that list here to include a further three (potentially) full species (*Cestum veneris*, *Beroe forskalii*?, *Ocyropsis maculata*?) and six morphospecies (*Hormiphora* sp., *Leucothea* sp., *Bolinopsis* sp., *Eurhamphaea* sp., *Coeloplana* sp., *Lyrocteis* sp.). Because most of the effort is focused on the nearshore environment, for obvious reasons, the species observed are those with an affinity for such environments –hence, with the exception of *Lyrocteis*, the absence of deeper water forms. The images of *Lyrocteis* shown here were collected from about 70 m depth using a remotely operated

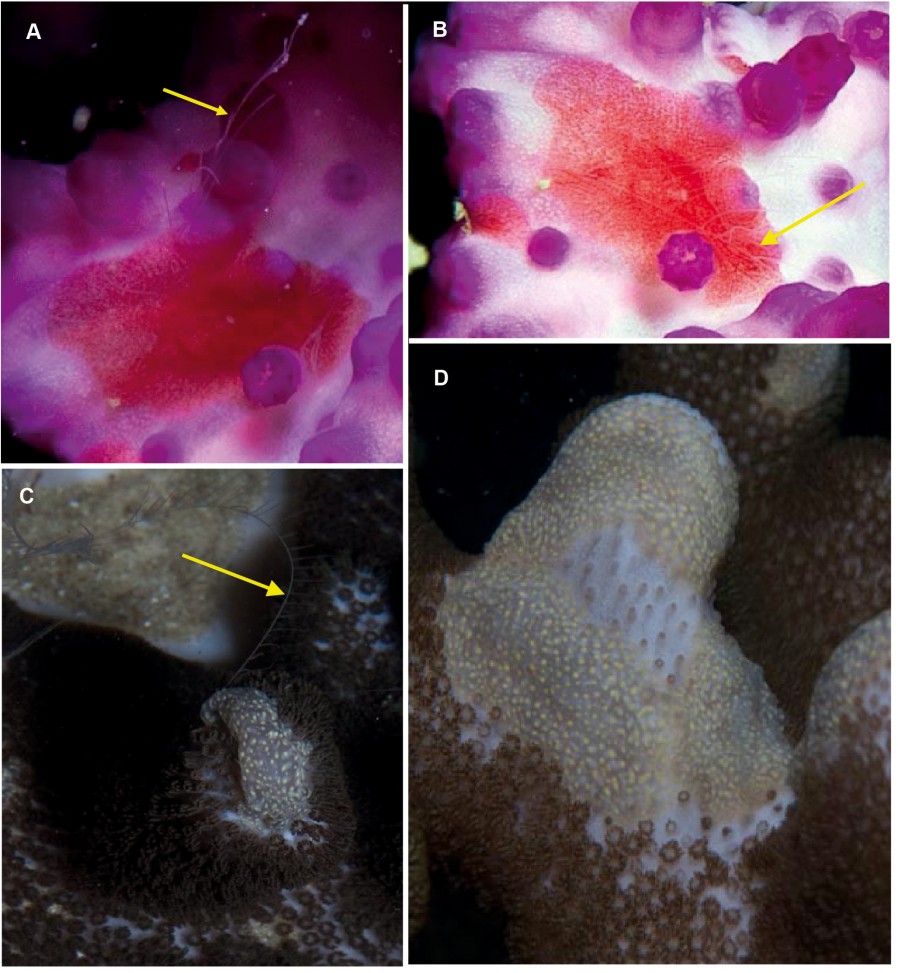

**Figure 11** *Coeloplana* **sp. Cryptic species of benthic ctenophores that lack ctene rows but possess tentacles (arrows).** A knowledge of the species on which specimens of *Coeloplana* are found can assist in identification. Photographs provided by: Georgina Jones; (A, B) taken in False Bay, (C, D) from southern Mozambique.

vehicle as part of ongoing investigations into the ecosystem occupied by the coelacanth *Latimeria chalumnae*.

The genera observed have widespread distributions in coastal areas (Table 1) and have been recorded off Australia (*Gershwin, Zeidler & Davie, 2010*), in the Mediterranean Sea (*Madin, 1991*; *Shiganova & Malej, 2009*; *Deidun, 2011*; *Çinar et al., 2014*), in the NW (*Mayer, 1912*) and NE Atlantic (*Moro et al., 2013*), around South America (*Oliveira et al., 2007*; *Oliveira et al., 2016*; *Schiariti et al., 2020*) and along the Pacific (*Wrobel & Mills, 1998*; *Mills & Haddock, 2007*; *Ruiz-Escobar, Valadez-Vargas & Oliveira, 2015*) and Atlantic (*Mayer, 1912*) coasts of North America. The taxa that would appear to be missing from regional waters include species of *Lampea* as well as *Deiopea kaloktenota* and *Velamen parallelum*, all of which are commonly reported from temperate and subtropical coastal environments elsewhere. Species of *Lampea* resemble those of the common cydippid

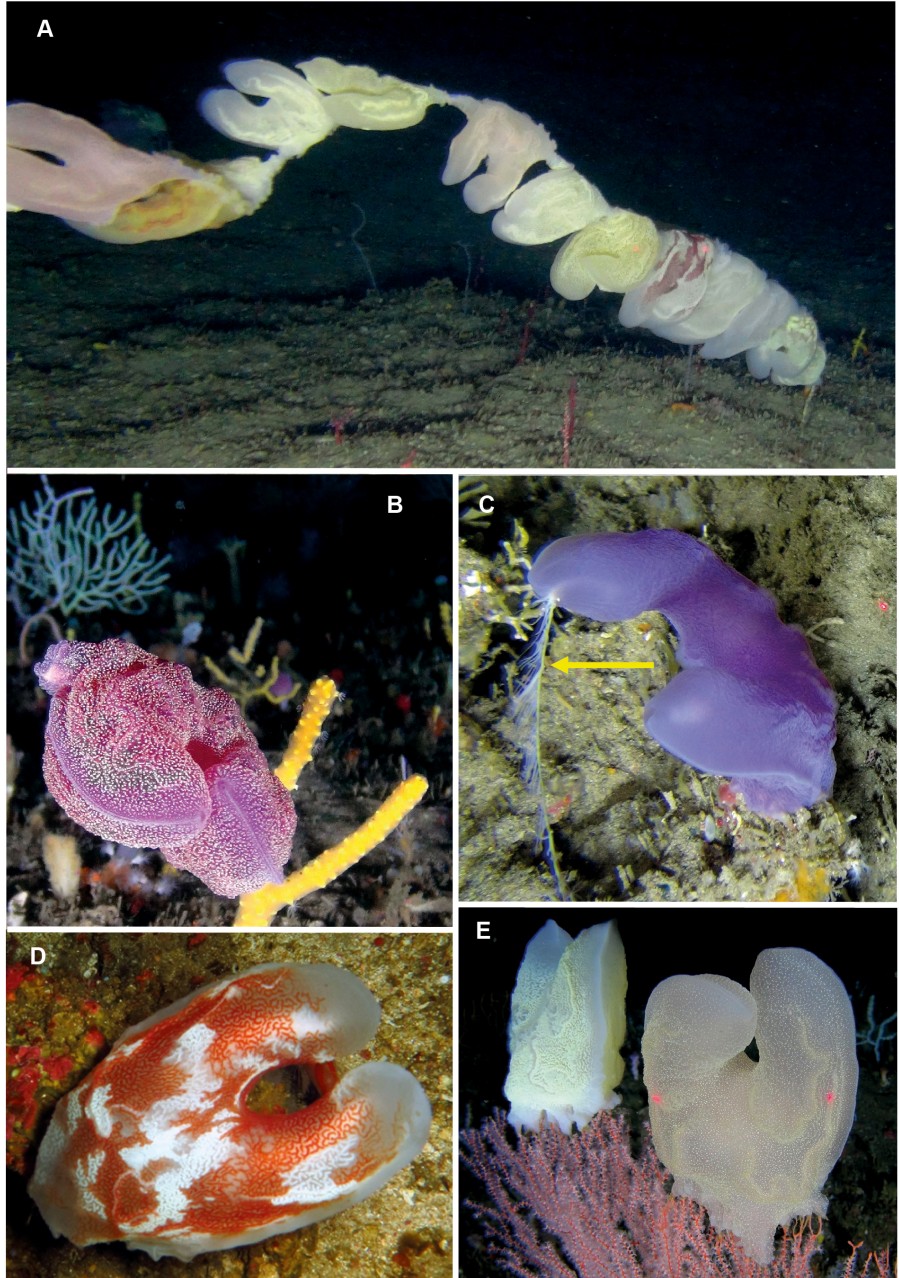

**Figure 12** *Lyrocteis* **sp. A lyre-shaped species of benthic ctenophore, without ctene rows as adult and with two tentacles (one indicated by arrow, C) that emerge from each of the two finger-like projections (A–E).** All specimens were photographed at depths of about 70 m using a remotely operated vehicle in the canyons of Sodwana Bay and ISimangaliso Wetland Park. Photographs provided by Ryan Palmer, using the research platform of the South African Institute of Aquatic Biodiversity (SAIAB) to the ACEP Surrogacy project, the ACEP Canyon Connections project, the ACEP Spatial Solutions project and the ACEP Protea Canyon project. ACEP Surrogacy project, the ACEP Canyon Connections project, the ACEP Spatial Solutions project and the ACEP Protea Canyon project.

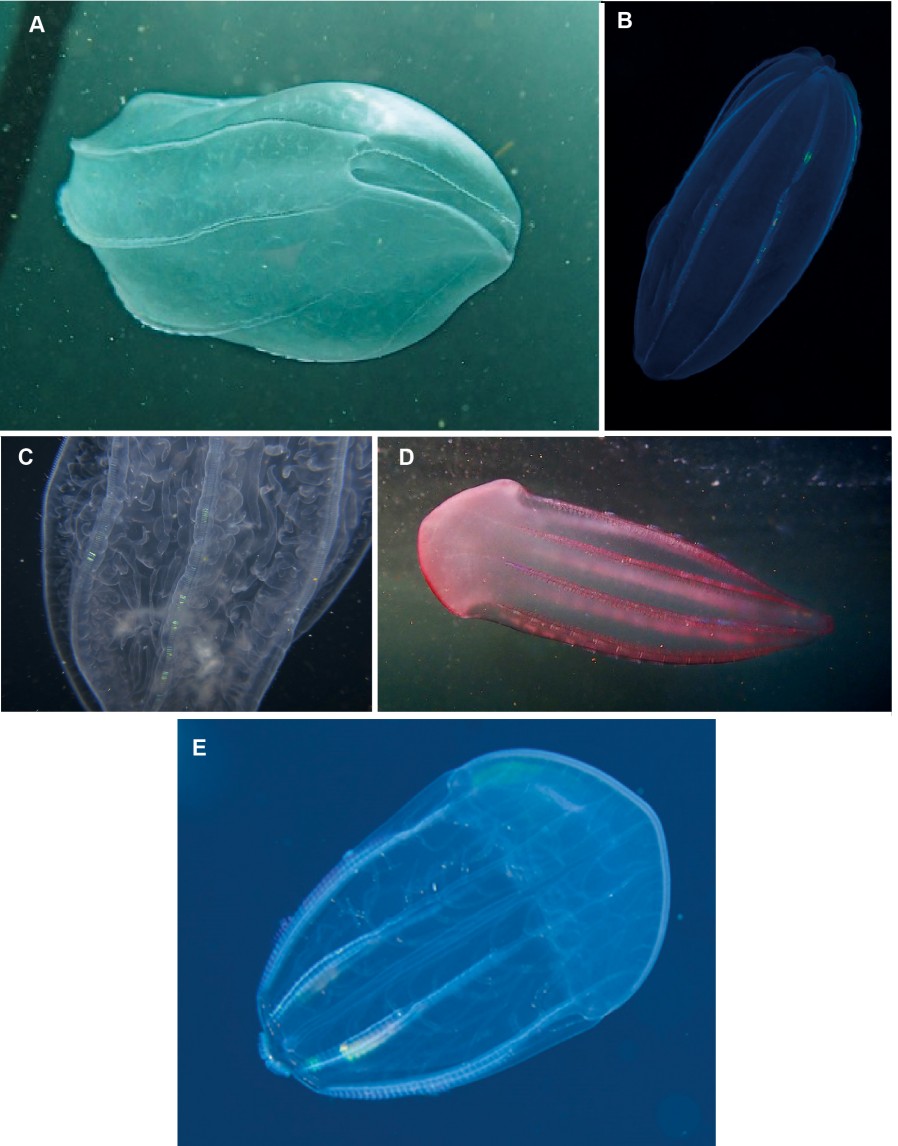

**Figure 13** ***Beroe* sp. (A–C), *Beroe forskalii*? (D) and *Beroe mitrata*? (E).** Species of *Beroe* lack tentacles and usually resemble cigars; the network of canals (C) in the body wall helps to separate the different species. *Beroe forskalii*? may be pink in colour and is markedly broader at the oral end. Photographs provided by: (A, D) Craig Foster, (B) Guido Zsilavecz, (C) Jenny Stromvoll, (E) Georgina Jones; (A, D) taken in False Bay, (B) at Oudekraal, and (C, E) in southern Mozambique.

*Pleurobrachia*, but have an avocado or bottle shape, not ovoid, with the body being drawn out orally; the ctene rows only extend to about half the body length and individuals have a large stomodeum. Species in this genus consume salps (*Harbison, Madin & Swanberg, 1978*), and given that large blooms of these can be encountered over the Agulhas Bank (*De Decker, 1973*) it is likely that *Lampea* is present in the region. Like species of *Lampea*, *Deiopea kaloktenota* rarely grows to more than 5 cm in size: it is very transparent, has short

lobes and has widely spaced ctenes on the ctene rows. Small size and fragility characterises the species that have not been recorded locally, which would agree with other community science-based efforts to map ctenophore diversity (e.g., *Deidun, 2011*).

Given South Africa's geographic position between the Indian and Atlantic oceans, and the strong nature of the Agulhas Current (*Lutjeharms, 2006*), the number of ctenophore species that can be expected is certain to be much higher than presently noted. Regional waters support between (approximately) 40–56% of the global holoplanktonic hydrozoan fauna (*Gibbons et al., 2010a*) and ~57% of the world's euphausiids (*Gibbons, Barange & Hutchings, 1995*), the latter being an holoplanktonic group of crustaceans. Whilst the majority of ctenophores are holoplanktonic and show limited endemism (but see *Gershwin, Zeidler & Davie, 2010*), some are benthic, and it is entirely likely that endemic species will be found amongst the Platyctenida (e.g., *Alamaru, Brokovich & Loya, 2015*). *Gibbons et al. (2010b)* noted that within Hydrozoa, those genera showing restricted patterns of dispersal tended to be more diverse than those with a relatively long-lived medusa phase. It comes as no surprise then that the two most specious ctenophore genera (*Coeloplana* and *Ctenoplana*) display similarly restricted patterns of dispersal.

Although a lesser number of ctenophores has been noted from around South Africa than Australia (*Gershwin, Zeidler & Davie, 2010*; Table 1), the length of the regional coastline is only about 3 700 km. The latter distance equates to about 50% of that of Brazil, which boasts 13 species: not morphospecies (*Oliveira et al., 2016*). Consequently, the diversity of ctenophores in the region must be regarded as comparatively high, especially given that there are no locally situated ctenophore taxonomists, like there are in North America (e.g., *Mills & Haddock, 2007*) or Brazil (e.g., *Oliveira et al., 2016*), and as there were in Australia (e.g., *Matsumoto & Gowlett-Holmes, 1996*; *Gershwin, Zeidler & Davie, 2010*). In these other environments, many species have only been observed through specialised and dedicated methods, such as submersibles and blue-water scuba diving, so additional diversity is sure to be uncovered over time. As has been suggested for other taxa, the high diversity noted in the region must be attributable to the diversity of water masses and environments that occur. Environments that span the gamut from coral reefs in the extreme NE of the country, which experience water temperatures in the upper 20s, to less than 8 °C noted in the kelp beds along the west coast (*Griffiths et al., 2010*).

The images shown here were solicited from the community scientists listed in the acknowledgements below. They represent the clearest, most unambiguous photographs reviewed and are the ones that we consider to be most suitable for illustrating the species recorded. It should be stressed, however, that these images were in the minority, and the vast majority of pictures were either snapped in turbid waters when animals were partially obscured by bubbles or detritus, or the specimen was moving and the images were slightly blurred and out of focus. A professional photographer, or even a selective amateur, is likely to quickly consign such images to "trash"—writing them off as embarrassing mistakes. But regardless of how embarrassing they may seem to the photographer, these pictures can still be valuable to science if the specimens they portray can be identified to morphospecies. Remember, mapping diversity is one thing, but platforms such as iNaturalist also allow us to understand ecological and environmental change, IF they are used by community scientists
**Table 1  List of the ctenophore genera that have been described or photographed around South Africa, with an indication of how many species are recorded in each.** Comparative information provided for other coastal regions around the world, based on recent regional treatments that have not been updated since their publication. *known introduction. Source—1 this study, 2 *Deidun (2011)*, 3 *Çinar et al. (2014)*, 4 *Shiganova & Malej (2009)*, 5 *Ruiz-Escobar, Valadez-Vargas & Oliveira (2015)*, 6 *Schiariti et al. (2020)*, 7 *Oliveira et al. (2007)*, 8 *Gershwin, Zeidler & Davie (2010)*, 9 *O'Sullivan (1986)*, 10 *Oliveira et al. (2016)*, 11 *Moro et al. (2013)*. Information on coastline length (km) obtained from https://en.wikipedia.org/wiki/List_of_countries_by_length_of_coastline.

| Region | *Hormiphora* | *Pleurobrachia* | *Callianira* | *Cestum* | *Ocyropsis* | *Bolinopsis* | *Leucothea* | *Eurhamphea* | *Coeloplana* | *Lyrocteis* | *Beroe* | Other taxa | Sum | Source | Coastline length (km) |
|---|---|---|---|---|---|---|---|---|---|---|---|---|---|---|---|
| South Africa | 1 | 2 | 1 | 1 | 1 | 1 | 1 | 1 | 1 | 1 | 4 | | 14 | 1 | 2 798 |
| Malta | | | | | | | 1 | | | | 2 | | 3 | 2 | 253 |
| Turkey | | 2 | | 1 | | 2 | | | | | 1* | *Mnemiopsis leidyi** | 5+1* | 3 | 7 200 |
| N Adriatic | 1 | 3 | | | | 1 | 1 | | | | 3+1* | *Haeckelia rubra, Lampea pancerina, Deiopea kaloktenota, Mnemiopsis leidyi** | 12+2* | 4 | 2 323 |
| W Mexico | 1 | 1 | | 1 | 2 | 1 | | | | | | *Velamen parallelum* | 7 | 5 | 7 338 |
| Argentina, Uruguay | | 1 | 1 | | | | | 1 | | | 3 | *Mnemiopsis leidyi, Lampea pancerina, Mertensia ovum* | 9 | 6 | 5 649 |
| Brazil | | | | 1 | 2 | 1 | 1 | 1 | | | 2 | *Lampea pancerina, Vallicula multiformis, Bolinopsis vitrea, Mnemiopsis leidyi, Velamen parallelum* | 13 | 7 | 7 491 |
| Australia | | 3 | | | 5 | 4 | 2 | | 8 | | 5 | *Euplokamis evansae, Pukia falcata, Velamen parallelum, Neis cordigera, Ctenoplana sp* | 32 | 8 | 25 760 |
| Antarctica | | 1 | 2 | | | | | | | 1 | 3 | *Cryptocoda gerlachi, Bathyctena chuni* | 7 | 9 | 17 968 |
| Peru | 1 | 1 | | | | | | | | | 2 | *Velamen parallelum* | 5 | 10 | 2 414 |
| Chile | | 1 | 1 | 1 | | | | | | | 2 | *Thallassocalyce inconstans, Mertensia ovum, Velamen parallelum* | 8 | 10 | 6 435 |
| Colombia | | | | 1 | | | | | | | 1 | *Mnemiopsis leidyi* | 3 | 10 | 3 208 |
| Canary Islands | 3 | 1 | 1 | 1 | 1 | 1 | 1 | 1 | | | 2 | *Velamen parallelum, Tinerfe cyanea, Charistephane fugiens, Vallicula multiformis* | 13 | 11 | 1 501 |
**Table 2 The total number of dives made by Peter Southwood in False Bay and environs over the period 2003–2019.** The number of separate dives each month in which different ctenophore morphospecies were photographed is also shown (percentage dives in parentheses).

| Month | Jan | Feb | Mar | Apr | May | Jun | Jul | Aug | Sep | Oct | Nov | Dec |
|---|---|---|---|---|---|---|---|---|---|---|---|---|
| Total No Dives | 68 | 55 | 58 | 56 | 47 | 52 | 65 | 43 | 55 | 51 | 65 | 96 |
| *Pleurobrachia* | 1 (1.5) | 2 (3.6) | | 1 (1.8) | 1 (2.1) | | 1 (1.5) | | | | | |
| *Hormiphora* | | | 1 (1.7) | | | | | | | 1 (2) | | |
| *Cestum* | | | | | | | | | | | 1 (1.5) | |
| *Ocyropsis* | | | | 2 (3.6) | 5 (10.6) | 3 (5.8) | | | | | | |
| *Leucothea* | | | | | 1 (2.1) | 4 (7.7) | 1 (1.5) | 2 (4.7) | 2 (3.6) | | | |
| *Bolinopsis* | | | 3 (5.2) | 2 (3.6) | 3 (6.4) | 2 (3.8) | 5 (7.7) | | 1 (1.8) | | | |
| *Beroe* | 6 (8.8) | 8 (14.5) | 3 (5.2) | 2 (3.6) | 2 (4.3) | | 5 (7.7) | 1 (2.3) | 3 (5.5) | 2 (3.9) | 1 (1.5) | 3 (3.1) |

on a routine and ongoing basis. We illustrate what we mean, using the photographs taken by Peter Southwood.

Peter is a retired engineer, and he has been taking underwater photographs in False Bay since 2003. He usually dives at least once every weekend when he is at home, and he has captured more than 120,000 digital images (508 GB) of marine animals that span the taxonomic gamut from sponges to dolphins. Table 2 displays the number of dives in which Peter has photographed ctenophores, by month over the period 2003–2019 ($n = 711$). These data suggest that, in False Bay, ctenophores are relatively uncommon and are seen only on about 11% of trips. *Beroe* can be found throughout the year, *Leucothea* is present from May to September, whilst *Ocyropsis* is only seen between April and June. These observations may be real and may impart real information about seasonal changes in distribution that we could try and relate to seasonal changes in the oceanographic environment within False Bay. However, they are also biased and will reflect Peter's willingness to hold on to poor images, his willingness to take a photograph of "yet another ctenophore" when the visibility is poor and he is running out of air. Consequently, scientists must be careful when they try and use such data in this way, unless community scientists routinely capture and share their photographs. It is a "numbers game": the more data collectors there are, the greater the chances that images not taken by Peter on any given day, will be captured by someone else.

## ACKNOWLEDGEMENTS

We dedicate this paper to all the different local community scientists that have contributed photographs for this project (Peter Southwood, Georgina Jones, Geoff Spiby, Mike and Val Fraser, Dennis King, Louis van Wyk, Jenny Stromvoll, Guido Zsilavecz), and we urge them to keep up their excellent efforts and to spread the word: all the photographs used here remain their property. Community science platforms such as iNaturalist (http://www.inaturalist.org) and Jellywatch (http://www.jellywatch.org) are to be applauded for the work they do, and we encourage all interested persons who have taken photographs of ctenophores, however embarrassing they may seem to you personally, to upload them to the iNaturalist site so we can push this and other biodiversity

projects forwards. It should be remembered that verified photographs are used by the AI algorithms in iNaturalist to assist community members in the identification of posted specimens. Thanks to Cathy Boucher for assistance with the preparation of the figures. This manuscript has benefited from the comments supplied by our colleague Larry Madin and another ''anonymous'' reviewer, and we are grateful for their efforts in this regard.

### Funding

This work was supported by the National Research Foundation and the University of the Western Cape. Additional support was provided by the ACEP Surrogacy project, the ACEP Canyon Connections project, the ACEP Spatial Solutions project and the ACEP Protea Canyon project. The funders had no role in study design, data collection and analysis, decision to publish, or preparation of the manuscript.

### Grant Disclosures

The following grant information was disclosed by the authors:
National Research Foundation.
University of the Western Cape.
ACEP Surrogacy project.
ACEP Canyon Connections project.
ACEP Spatial Solutions project.
ACEP Protea Canyon project.

### Competing Interests

Craig Foster is the co founder of the Sea Change Trust, which ''works closely with local government, major academic institutions, influencers and strategic partners who share our vision to protect one of Earth's last wild places: the South African kelp forest.''

The authors declare there are no competing interests.

### Author Contributions

- Mark J. Gibbons conceived and designed the experiments, performed the experiments, analyzed the data, prepared figures and/or tables, authored or reviewed drafts of the paper, and approved the final draft.
- Steve H.D. Haddock and George I. Matsumoto performed the experiments, analyzed the data, authored or reviewed drafts of the paper, and approved the final draft.
- Craig Foster performed the experiments, prepared figures and/or tables, authored or reviewed drafts of the paper, and approved the final draft.

### Field Study Permissions

The following information was supplied relating to field study approvals (i.e., approving body and any reference numbers):

All the photographs used here were taken by community scientists at a variety of open locations around South Africa: no specific permits are required to take underwater photographs of animals around South Africa.

## Data Availability

The photographs used as primary data in this article are made available as Figs. 1–13.

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
