# Peer review of "Records of ctenophores from South Africa"

_PeerJ, doi:10.7717/peerj.10697_

## Round 0.1 · original submission · Minor Revisions

Dear Mark,
Thank you for the interesting paper. I eventually got two referees reports with some constructive suggestions. The idea of posting photos to iNaturalist is a good one because verified photos are used to train its AI and thus help identify photos posted by citizens. While this is not required for the paper to be published it is a form of 'making primary materials' available to researchers you may consider. [I use iNaturalist myself a lot]

Reviewer 1 ·

Basic reporting

Overall, the article is well-written and does a great job of relaying a great deal of information in an accessible way. It’s easy to imagine this being a helpful resource for people at many levels of scientific experience and interest. That said, there are a few points I would like to see the authors address that I think could improve its ease of use.

First, the article should have a Materials and Methods section. I realize that as a citizen science-based biodiversity project, the composition of such a section might be somewhat atypical, nonetheless I think it would be helpful for readers. I provide some suggestions for topics that might be helpful for such a section to cover in section 2 of this review.

Also, the clarity of the photographic advice would be improved if it were set off into its own (sub)section, perhaps with a bulleted list of key points. It might also be worth distinguishing advice for photographing ctenophores in the ocean in a SCUBA context vs photographing sampled ctenophores in holding tanks/aquaria.

Experimental design

The topic of the article is interesting from a biodiversity perspective, and it fits within PeerJ's aims and scope.

However, the article needs a materials and methods section describing how the data were collected in more detail. Some points of possible interest for such a section,

• Examples of types of cameras/gear used by citizen scientists in this study
• Was there a standard protocol used by citizen scientists used to record environmental data, location, etc?
• Did citizen scientists use a web portal (such as iNaturalist) to collect/organize photos/metadata or did they correspond directly with organizers?
• Were citizen scientists recruited specifically to help describe ctenophore biodiversity or were they participating in other, broader efforts?

Validity of the findings

The findings of the article seem valid and well-supported overall, but I have a few points of clarification,

Are there any additional photographs of ctenophores from the citizen scientists that exist but didn’t fit into the present article? If so and if they are on an online repository such as iNaturalist it would be worthwhile to provide a link to the relevant user profiles so readers can access that content.

Similarly, do more detailed data on locations/timing for observed ctenophores exist? If so, it might be helpful/worthwhile to include/discuss it, even if it is somewhat limited. It might help future citizen scientists in the area know how far out from shore to direct observations, when to time sampling relative to the tides, etc.

Additional comments

Line 77-78: Please cite a basis for this statement and be specific in the claim (e.g. "based on the molecular phylogenies of studies X and Y the affinities of ctenophores and cnidarians appear convergent and not based on shared ancestry")

The section on the ctenophore body plan should make explicit mention of the term biradial symmetry.

Line 294: Eurhamphaea sp. aboral processes/filaments should not be referred to as “tentacles” in order to distinguish them from feeding structures. I think the language of Mayer, 1912 is generally a good guide, “apical processes […] give rise to a pair of […] simple filaments” (p. 41). A similar change should be made in the caption for figure 11. Also in the caption for figure 11, I would say that the apical filaments are not a great diagnostic for this genus, as the lengths can vary quite a bit, and in some individuals, the filaments may be so reduced that they are hard to even see. It would probably be better to mention the ink spots in this caption.

Figures 4 and 5: Are these meant to be one figure with two panels, a and b? If not, neither needs a panel label.

Figure 13 caption: Which photo shows a species likely new to science? Panels a, b, d, and e should also be mentioned in the caption.

Double check citations. There are some spelling discrepancies and some citations that were cited in the text but not present in the reference section, e.g.

Line 80: Giribet and Edgecombe, 2020 not in bibliography
Line 128: Should read Deidun
Line 333: Should read Moser (1910)

Finally, if there’s any way to show the approximate observed sightings or ranges for the different taxa discussed on a map of SA, I think such a figure would add a lot to the article.

·

Basic reporting

This ms combines descriptive, taxonomic and ecological information in a report on the occurrence of ctenophore species of South African coastal waters. Unlike more conventional reports, it is based on photographs of ctenophores from citizen scientists, along with prior published information on ctenophore occurrence and systematics.
The underlying data of the paper are thus the photographic images, which have been identified by the authors, at least two of whom (Haddock, Matsumoto) are well known experts in ctenophore biology.
The authors emphasize that the delicacy of ctenophores and the difficulty of preserving specimens makes reliance on images unavoidable as a basis for descriptive studies. Presumably collection of images rather than specimens obviates any concern about proper permitting.
The Introduction established the physical context of the study region, including the influence of the Agulhas Current and the gradients in temperature, productivity and biomass that result in different parts of the South African coast.
There are paragraphs describing general ecology of ctenophores, including impacts as introduced species, and evolutionary history. A brief section on photographing ctenophores seems aimed at citizen scientist and divers who may wish to contribute future images.
A few specific comments:
Abstract: Authors say only six species are previously described from South Africa, and that they add three more at species level and 6 at morphospecies (=genus here). But I don’t see anywhere an explicit notation of which are which in the subsequent descriptions.
Line 55: ‘..some are planktonic others are strictly benthic..’ Really, the large majority of species are planktonic, so better to say ‘most are..’
Line 59: ‘..ensnaring prey in mucous’. Although mucus is used by lobates and cestids to trap and transport prey, the colloblasts themselves use a glue to stick to prey, not really mucus.
Line 80: reference Giribet & Edgecombe 2020 isn’t in the references.
Line 104: Since this paper seems aimed at non-specialists on ctenophores, a definition of ‘morphospecies’ here would be useful.
Lines 106-124: These paragraphs on photographic hints should be set off with a subheading and placed at the end of this section, before the overview of classification and anatomy.
The Overview is a good addition here and should be useful to citizen scientists to understand ctenophore morphology. Figures are helpful.

Experimental design

There is no 'experiment' in this paper. It is a descriptive and taxonomic report. These comments refer to aspects of the data presentation.

Descriptive Section

The subhead here is ‘Morphospecies’ but some are described at species level and others at genus level. Previously 3 new ‘fully described’ and 6 morphospecies were mentioned, but there seem to be 11 separate species/morphospecies in this section. This should be clarified.
The descriptions all seem accurate and clear. Presumably they will have the relevant Figures adjacent, since images are really the best way to identify ctenophores to this level.

Figure 4: the caption says Hormiphora sp., but the image and link are Pleurobrachia pileus.
Discussion
Line 358: interesting that Lampea was not found, as it preys on salps, which are often abundant in South African waters.

Validity of the findings

Conclusions
The authors cite the value of image-based data for description of morphospecies, which is sufficient to understand the ecological role and importance of the ctenophores present. This seems a reasonable perspective.

General Comments:
Even if not a rigorous taxonomic treatment, this paper provides usefully accurate new information about the presence and distribution of several new ctenophore species. Based on amateur contributions of photographic data and directed in part to that community, the paper helps fill gaps in knowledge of the plankton fauna of the region, and encourages further citizen participation in making and reporting observations. The authors all have relevant expertise and reputations in zooplankton and ctenophore biology.

Additional comments

see above

---

## Round 0.2 · accepted · Accept

Thank you for the thorough and thoughtful amendments to the paper. The photographs are stunning and a highlight. I am sure they will be useful to others.